# MIXOUT: EFFECTIVE REGULARIZATION TO FINETUNE LARGE-SCALE PRETRAINED LANGUAGE MODELS

**Cheolhyoung Lee**[*]
cheolhyoung.lee@kaist.ac.kr

**Kyunghyun Cho**[† ‡ §]
kyunghyun.cho@nyu.edu

**Wanmo Kang**[*]
wanmo.kang@kaist.ac.kr

## ABSTRACT

In natural language processing, it has been observed recently that generalization could be greatly improved by finetuning a large-scale language model pretrained on a large unlabeled corpus. Despite its recent success and wide adoption, finetuning a large pretrained language model on a downstream task is prone to degenerate performance when there are only a small number of training instances available. In this paper, we introduce a new regularization technique, to which we refer as "mixout", motivated by dropout. Mixout stochastically mixes the parameters of two models. We show that our mixout technique regularizes learning to minimize the deviation from one of the two models and that the strength of regularization adapts along the optimization trajectory. We empirically evaluate the proposed mixout and its variants on finetuning a pretrained language model on downstream tasks. More specifically, we demonstrate that the stability of finetuning and the average accuracy greatly increase when we use the proposed approach to regularize finetuning of BERT on downstream tasks in GLUE.

## 1 INTRODUCTION

Transfer learning has been widely used for the tasks in natural language processing (NLP) (Collobert et al., 2011; Devlin et al., 2018; Yang et al., 2019; Liu et al., 2019; Phang et al., 2018). In particular, Devlin et al. (2018) recently demonstrated the effectiveness of finetuning a large-scale language model pretrained on a large, unannotated corpus on a wide range of NLP tasks including question answering and language inference. They have designed two variants of models, $\text{BERT}_{\text{LARGE}}$ (340M parameters) and $\text{BERT}_{\text{BASE}}$ (110M parameters). Although $\text{BERT}_{\text{LARGE}}$ outperforms $\text{BERT}_{\text{BASE}}$ generally, it was observed that finetuning sometimes fails when a target dataset has fewer than 10,000 training instances (Devlin et al., 2018; Phang et al., 2018).

When finetuning a big, pretrained language model, dropout (Srivastava et al., 2014) has been used as a regularization technique to prevent co-adaptation of neurons (Vaswani et al., 2017; Devlin et al., 2018; Yang et al., 2019). We provide a theoretical understanding of dropout and its variants, such as Gaussian dropout (Wang & Manning, 2013), variational dropout (Kingma et al., 2015), and dropconnect (Wan et al., 2013), as an adaptive $L^2$-penalty toward the origin (all zero parameters $\mathbf{0}$) and generalize dropout by considering a target model parameter $\boldsymbol{u}$ (instead of the origin), to which we refer as $\texttt{mixout}(\boldsymbol{u})$. We illustrate $\texttt{mixout}(\boldsymbol{u})$ in Figure 1. To be specific, $\texttt{mixout}(\boldsymbol{u})$ replaces all outgoing parameters from a randomly selected neuron to the corresponding parameters of $\boldsymbol{u}$. $\texttt{mixout}(\boldsymbol{u})$ avoids optimization from diverging away from $\boldsymbol{u}$ through an adaptive $L^2$-penalty toward $\boldsymbol{u}$. Unlike $\texttt{mixout}(\boldsymbol{u})$, dropout encourages a move toward the origin which deviates away from $\boldsymbol{u}$ since dropout is equivalent to $\texttt{mixout}(\mathbf{0})$.

We conduct experiments empirically validating the effectiveness of the proposed $\texttt{mixout}(\boldsymbol{w}_{\text{pre}})$ where $\boldsymbol{w}_{\text{pre}}$ denotes a pretrained model parameter. To validate our theoretical findings, we train a fully connected network on EMNIST Digits (Cohen et al., 2017) and finetune it on MNIST. We observe that a finetuning solution of $\texttt{mixout}(\boldsymbol{w}_{\text{pre}})$ deviates less from $\boldsymbol{w}_{\text{pre}}$ in the $L^2$-sense than

---

[*]Department of Mathematical Sciences, KAIST, Daejeon, 34141, Republic of Korea

[†]New York University

[‡]Facebook AI Research

[§]CIFAR Azrieli Global Scholar

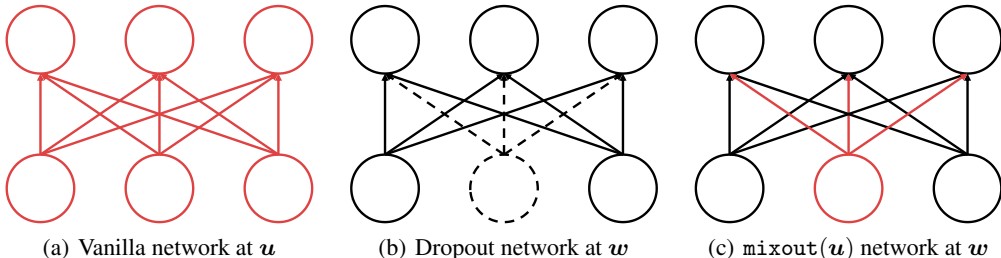

(a) Vanilla network at $\boldsymbol{u}$        (b) Dropout network at $\boldsymbol{w}$        (c) `mixout`$(\boldsymbol{u})$ network at $\boldsymbol{w}$

Figure 1: Illustration of `mixout`$(\boldsymbol{u})$. Suppose that $\boldsymbol{u}$ and $\boldsymbol{w}$ are a target model parameter and a current model parameter, respectively. (a): We first memorize the parameters of the vanilla network at $\boldsymbol{u}$. (b): In the dropout network, we randomly choose an input neuron to be dropped (a dotted neuron) with a probability of $p$. That is, all outgoing parameters from the dropped neuron are eliminated (dotted connections). (c): In the `mixout`$(\boldsymbol{u})$ network, the eliminated parameters in (b) are replaced by the corresponding parameters in (a). In other words, the `mixout`$(\boldsymbol{u})$ network at $\boldsymbol{w}$ is the mixture of the vanilla network at $\boldsymbol{u}$ and the dropout network at $\boldsymbol{w}$ with a probability of $p$.

that of dropout. In the main experiment, we finetune $\text{BERT}_{\text{LARGE}}$ with `mixout`$(\boldsymbol{w}_{\text{pre}})$ on small training sets of GLUE (Wang et al., 2018). We observe that `mixout`$(\boldsymbol{w}_{\text{pre}})$ reduces the number of unusable models that fail with the chance-level accuracy and increases the average development (dev) scores for all tasks. In the ablation studies, we perform the following three experiments for finetuning $\text{BERT}_{\text{LARGE}}$ with `mixout`$(\boldsymbol{w}_{\text{pre}})$: (i) the effect of `mixout`$(\boldsymbol{w}_{\text{pre}})$ on a sufficient number of training examples, (ii) the effect of a regularization technique for an additional output layer which is not pretrained, and (iii) the effect of probability of `mixout`$(\boldsymbol{w}_{\text{pre}})$ compared to dropout. From these ablation studies, we observe that three characteristics of `mixout`$(\boldsymbol{w}_{\text{pre}})$: (i) finetuning with `mixout`$(\boldsymbol{w}_{\text{pre}})$ does not harm model performance even with a sufficient number of training examples; (ii) It is beneficial to use a variant of mixout as a regularization technique for the additional output layer; (iii) The proposed `mixout`$(\boldsymbol{w}_{\text{pre}})$ is helpful to the average dev score and to the finetuning stability in a wider range of its hyperparameter $p$ than dropout.

## 1.1 RELATED WORK

For large-scale pretrained language models (Vaswani et al., 2017; Devlin et al., 2018; Yang et al., 2019), dropout has been used as one of several regularization techniques. The theoretical analysis for dropout as an $L^2$-regularizer toward $\boldsymbol{0}$ was explored by Wan et al. (2013) where $\boldsymbol{0}$ is the origin. They provided a sharp characterization of dropout for a simplified setting (generalized linear model). Mianjy & Arora (2019) gave a formal and complete characterization of dropout in deep linear networks with squared loss as a nuclear norm regularization toward $\boldsymbol{0}$. However, neither Wan et al. (2013) nor Mianjy & Arora (2019) gives theoretical analysis for the extension of dropout which uses a point other than $\boldsymbol{0}$.

Wiese et al. (2017), Kirkpatrick et al. (2017), and Schwarz et al. (2018) used $L^2$-penalty toward a pretrained model parameter to improve performance. They focused on preventing catastrophic forgetting to enable their models to learn multiple tasks sequentially. They however do not discuss nor demonstrate the effect of $L^2$-penalty toward the pretrained model parameter on the stability of finetuning. Barone et al. (2017) introduced tuneout, which is a special case of mixout. They applied various regularization techniques including dropout, tuneout, and $L^2$-penalty toward a pretrained model parameter to finetune neural machine translation. They however do not demonstrate empirical significance of tuneout compared to other regularization techniques nor its theoretical justification.

## 2 PRELIMINARIES AND NOTATIONS

**Norms and Loss Functions** Unless explicitly stated, a norm $\|\cdot\|$ refers to $L^2$-norm. A loss function of a neural network is written as $\mathcal{L}(\boldsymbol{w}) = \frac{1}{n}\sum_{i=1}^{n}\mathcal{L}_i(\boldsymbol{w})$, where $\boldsymbol{w}$ is a trainable model parameter. $\mathcal{L}_i$ is "a per-example loss function" computed on the $i$-th data point.

**Strong Convexity**  A differentiable function $f$ is strongly convex if there exists $m > 0$ such that

$$f(\boldsymbol{y}) \geq f(\boldsymbol{x}) + \nabla f(\boldsymbol{x})^\top (\boldsymbol{y} - \boldsymbol{x}) + \frac{m}{2} \|\boldsymbol{y} - \boldsymbol{x}\|^2, \tag{1}$$

for all $\boldsymbol{x}$ and $\boldsymbol{y}$.

**Weight Decay**  We refer as "wdecay($\boldsymbol{u}$, $\lambda$)" to minimizing

$$\mathcal{L}(\boldsymbol{w}) + \frac{\lambda}{2} \|\boldsymbol{w} - \boldsymbol{u}\|^2,$$

instead of the original loss function $\mathcal{L}(\boldsymbol{w})$ where $\lambda$ is a regularization coefficient. Usual weight decay of $\lambda$ is equivalent to wdecay($\boldsymbol{0}$, $\lambda$).

**Probability for Dropout and Dropconnect**  Dropout (Srivastava et al., 2014) is a regularization technique selecting a neuron to drop with a probability of $p$. Dropconnect (Wan et al., 2013) chooses a parameter to drop with a probability of $p$. To emphasize their hyperparameter $p$, we write dropout and dropconnect with a drop probability of $p$ as "dropout($p$)" and "dropconnect($p$)", respectively. dropout($p$) is a special case of dropconnect($p$) if we simultaneously drop the parameters outgoing from each dropped neuron.

**Inverted Dropout and Dropconnect**  In the case of dropout($p$), a neuron is retained with a probability of $1-p$ during training. If we denote the weight parameter of that neuron as $\boldsymbol{w}$ during training, then we use $(1 - p)\boldsymbol{w}$ for that weight parameter at test time (Srivastava et al., 2014). This ensures that the expected output of a neuron is the same as the actual output at test time. In this paper, dropout($p$) refers to inverted dropout($p$) which uses $\boldsymbol{w}/(1 - p)$ instead of $\boldsymbol{w}$ during training. By doing so, we do not need to compute the output separately at test time. Similarly, dropconnect($p$) refers to inverted dropconnect($p$).

## 3  ANALYSIS OF DROPOUT AND ITS GENERALIZATION

We start our theoretical analysis by investigating dropconnect which is a general form of dropout and then apply the result derived from dropconnect to dropout. The iterative SGD equation for dropconnect($p$) with a learning rate of $\eta$ is

$$\boldsymbol{w}^{(t+1)} = \boldsymbol{w}^{(t)} - \eta \boldsymbol{B}^{(t)} \nabla \mathcal{L}\left( \left(\mathbb{E}B_1^{(t)}\right)^{-1} \boldsymbol{B}^{(t)} \boldsymbol{w}^{(t)} \right), \quad t = 0,\ 1,\ 2,\ \cdots, \tag{2}$$

where $\boldsymbol{B}^{(t)} = \mathrm{diag}(B_1^{(t)},\ B_2^{(t)},\ \cdots,\ B_d^{(t)})$ and $B_i^{(t)}$'s are mutually independent Bernoulli$(1 - p)$ random variables with a drop probability of $p$ for all $i$ and $t$. We regard equation 2 as finding a solution to the minimization problem below:

$$\min_{\boldsymbol{w}} \mathbb{E}\mathcal{L}\left( (\mathbb{E}B_1)^{-1} \boldsymbol{B} \boldsymbol{w} \right), \tag{3}$$

where $\boldsymbol{B} = \mathrm{diag}(B_1,\ B_2,\ \cdots,\ B_d)$ and $B_i$'s are mutually independent Bernoulli$(1 - p)$ random variables with a drop probability of $p$ for all $i$.

Gaussian dropout (Wang & Manning, 2013) and variational dropout (Kingma et al., 2015) use other random masks to improve dropout rather than Bernoulli random masks. To explain these variants of dropout as well, we set a random mask matrix $\boldsymbol{M} = \mathrm{diag}(M_1,\ M_2,\ \cdots,\ M_d)$ to satisfy $\mathbb{E}M_i = \mu$ and $\mathrm{Var}(M_i) = \sigma^2$ for all $i$. Now we define a random mixture function with respect to $\boldsymbol{w}$ from $\boldsymbol{u}$ and $\boldsymbol{M}$ as

$$\Phi(\boldsymbol{w};\ \boldsymbol{u}, \boldsymbol{M}) = \mu^{-1}\big((\boldsymbol{I} - \boldsymbol{M})\boldsymbol{u} + \boldsymbol{M}\boldsymbol{w} - (1 - \mu)\boldsymbol{u}\big), \tag{4}$$

and a minimization problem with "mixconnect($\boldsymbol{u}$, $\mu$, $\sigma^2$)" as

$$\min_{\boldsymbol{w}} \mathbb{E}\mathcal{L}\big(\Phi(\boldsymbol{w};\ \boldsymbol{u}, \boldsymbol{M})\big). \tag{5}$$

We can view dropconnect($p$) equation 3 as a special case of equation 5 where $\boldsymbol{u} = \boldsymbol{0}$ and $\boldsymbol{M} = \boldsymbol{B}$. We investigate how mixconnect($\boldsymbol{u}$, $\mu$, $\sigma^2$) differs from the vanilla minimization problem

$$\min_{\boldsymbol{w}} \mathbb{E}\mathcal{L}(\boldsymbol{w}). \tag{6}$$

If the loss function $\mathcal{L}$ is strongly convex, we can derive a lower bound of $\mathbb{E}\mathcal{L}\big(\Phi(\boldsymbol{w};\ \boldsymbol{u}, \boldsymbol{M})\big)$ as in Theorem 1:

**Theorem 1.** *Assume that the loss function $\mathcal{L}$ is strongly convex. Suppose that a random mixture function with respect to $\boldsymbol{w}$ from $\boldsymbol{u}$ and $\boldsymbol{M}$ is given by $\Phi(\boldsymbol{w}; \boldsymbol{u}, \boldsymbol{M})$ in equation 4 where $\boldsymbol{M}$ is $\mathrm{diag}(M_1, M_2, \cdots, M_d)$ satisfying $\mathbb{E}M_i = \mu$ and $\mathrm{Var}(M_i) = \sigma^2$ for all $i$. Then, there exists $m > 0$ such that*

$$\mathbb{E}\mathcal{L}\big(\Phi(\boldsymbol{w}; \boldsymbol{u}, \boldsymbol{M})\big) \geq \mathcal{L}(\boldsymbol{w}) + \frac{m\sigma^2}{2\mu^2}\|\boldsymbol{w} - \boldsymbol{u}\|^2, \tag{7}$$

*for all $\boldsymbol{w}$ (Proof in Supplement A).*

Theorem 1 shows that minimizing the l.h.s. of equation 7 minimizes the r.h.s. of equation 7 when the r.h.s. is a sharp lower limit of the l.h.s. The strong convexity of $\mathcal{L}$ means that $\mathcal{L}$ is bounded from below by a quadratic function, and the inequality of equation 7 comes from the strong convexity. Hence, the equality holds if $\mathcal{L}$ is quadratic, and $\texttt{mixconnect}(\boldsymbol{u}, \mu, \sigma^2)$ is an $L^2$-regularizer with a regularization coefficient of $m\sigma^2/\mu^2$.

## 3.1 MIXCONNECT TO MIXOUT

We propose mixout as a special case of mixconnect, which is motivated by the relationship between dropout and dropconnect. We assume that

$$\boldsymbol{w} = \left(w_1^{(N_1)}, \cdots, w_{d_1}^{(N_1)}, w_1^{(N_2)}, \cdots, w_{d_2}^{(N_2)}, \cdots\cdots, w_1^{(N_k)}, \cdots, w_{d_k}^{(N_k)}\right),$$

where $w_j^{(N_i)}$ is the $j$th parameter outgoing from the neuron $N_i$. We set the corresponding $\boldsymbol{M}$ to

$$\boldsymbol{M} = \mathrm{diag}\left(M^{(N_1)}, \cdots, M^{(N_1)}, M^{(N_2)}, \cdots, M^{(N_2)}, \cdots\cdots, M^{(N_k)}, \cdots, M^{(N_k)}\right), \tag{8}$$

where $\mathbb{E}M^{(N_i)} = \mu$ and $\mathrm{Var}(M^{(N_i)}) = \sigma^2$ for all $i$. In this paper, we set $M^{(N_i)}$ to $\mathrm{Bernoulli}(1-p)$ for all $i$ and $\texttt{mixout}(\boldsymbol{u})$ hereafter refers to this correlated version of mixconnect with Bernoulli random masks. We write it as "$\texttt{mixout}(\boldsymbol{u}, p)$" when we emphasize the mix probability $p$.

**Corollary 1.1.** *Assume that the loss function $\mathcal{L}$ is strongly convex. We denote the random mixture function of $\texttt{mixout}(\boldsymbol{u}, p)$, which is equivalent to that of $\texttt{mixconnect}(\boldsymbol{u}, 1-p, p-p^2)$, as $\Phi(\boldsymbol{w}; \boldsymbol{u}, \boldsymbol{M})$ where $\boldsymbol{M}$ is defined in equation 8. Then, there exists $m > 0$ such that*

$$\mathbb{E}\mathcal{L}\big(\Phi(\boldsymbol{w}; \boldsymbol{u}, \boldsymbol{B})\big) \geq \mathcal{L}(\boldsymbol{w}) + \frac{mp}{2(1-p)}\|\boldsymbol{w} - \boldsymbol{u}\|^2, \tag{9}$$

*for all $\boldsymbol{w}$.*

Corollary 1.1 is a straightforward result from Theorem 1. As the mix probability $p$ in equation 9 increases to 1, the $L^2$-regularization coefficient of $mp/(1-p)$ increases to infinity. It means that $p$ of $\texttt{mixout}(\boldsymbol{u}, p)$ can adjust the strength of $L^2$-penalty toward $\boldsymbol{u}$ in optimization. $\texttt{mixout}(\boldsymbol{u})$ differs from $\mathrm{wdecay}(\boldsymbol{u})$ since the regularization coefficient of $\texttt{mixout}(\boldsymbol{u})$ depends on $m$ determined by the current model parameter $\boldsymbol{w}$. $\texttt{mixout}(\boldsymbol{u}, p)$ indeed regularizes learning to minimize the deviation from $\boldsymbol{u}$. We validate this by performing least squares regression in Supplement D.

We often apply dropout to specific layers. For instance, Simonyan & Zisserman (2014) applied dropout to fully connected layers only. We generalize Theorem 1 to the case in which mixout is only applied to specific layers, and it can be done by constructing $\boldsymbol{M}$ in a particular way. We demonstrate this approach in Supplement B and show that mixout for specific layers adaptively $L^2$-penalizes their parameters.

## 3.2 MIXOUT FOR PRETRAINED MODELS

Hoffer et al. (2017) have empirically shown that

$$\|\boldsymbol{w}_t - \boldsymbol{w}_0\| \sim \log t, \tag{10}$$

where $\boldsymbol{w}_t$ is a model parameter after the $t$-th SGD step. When training from scratch, we usually sample an initial model parameter $\boldsymbol{w}_0$ from a normal/uniform distribution with mean 0 and small variance. Since $\boldsymbol{w}_0$ is close to the origin, $\boldsymbol{w}_t$ is away from the origin only with a large $t$ by equation 10. When finetuning, we initialize our model parameter from a pretrained model parameter $\boldsymbol{w}_{\mathrm{pre}}$. Since we usually obtain $\boldsymbol{w}_{\mathrm{pre}}$ by training from scratch on a large pretraining dataset, $\boldsymbol{w}_{\mathrm{pre}}$ is often far away from the origin. By Corollary 1.1, dropout $L^2$-penalizes the model parameter for deviating away from the origin rather than $\boldsymbol{w}_{\mathrm{pre}}$. To explicitly prevent the deviation from $\boldsymbol{w}_{\mathrm{pre}}$, we instead propose to use $\texttt{mixout}(\boldsymbol{w}_{\mathrm{pre}})$.

# 4 VERIFICATION OF THEORETICAL RESULTS FOR MIXOUT ON MNIST

Wiese et al. (2017) have highlighted that $\text{wdecay}(\boldsymbol{w}_{\text{pre}})$ is an effective regularization technique to avoid catastrophic forgetting during finetuning. Because $\text{mixout}(\boldsymbol{w}_{\text{pre}})$ keeps the finetuned model to stay in the vicinity of the pretrained model similarly to $\text{wdecay}(\boldsymbol{w}_{\text{pre}})$, we suspect that the proposed $\text{mixout}(\boldsymbol{w}_{\text{pre}})$ has a similar effect of alleviating the issue of catastrophic forgetting. To empirically verify this claim, we pretrain a 784-300-100-10 fully-connected network on EMNIST Digits (Cohen et al., 2017), and finetune it on MNIST. For more detailed description of the model architecture and datasets, see Supplement C.1.

In the pretraining stage, we run five random experiments with a batch size of 32 for $\{1, 2, \cdots, 20\}$ training epochs. We use Adam (Kingma & Ba, 2014) with a learning rate of $10^{-4}$, $\beta_1 = 0.9$, $\beta_2 = 0.999$, $\text{wdecay}(\boldsymbol{0}, 0.01)$, learning rate warm-up over the first 10% steps of the total steps, and linear decay of the learning rate after the warm-up. We use $\text{dropout}(0.1)$ for all layers except the input and output layers. We select $\boldsymbol{w}_{\text{pre}}$ whose validation accuracy on EMNIST Digits is best (0.992) in all experiments.

For finetuning, most of the model hyperparameters are kept same as in pretraining, with the exception of the learning rate, number of training epochs, and regularization techniques. We train with a learning rate of $5 \times 10^{-5}$ for 5 training epochs. We replace $\text{dropout}(p)$ with $\text{mixout}(\boldsymbol{w}_{\text{pre}}, p)$. We do not use any other regularization technique such as $\text{wdecay}(\boldsymbol{0})$ and $\text{wdecay}(\boldsymbol{w}_{\text{pre}})$. We monitor $\|\boldsymbol{w}_{\text{ft}} - \boldsymbol{w}_{\text{pre}}\|^2$,[1] validation accuracy on MNIST, and validation accuracy on EMNIST Digits to compare $\text{mixout}(\boldsymbol{w}_{\text{pre}}, p)$ to $\text{dropout}(p)$ across 10 random restarts.[2]

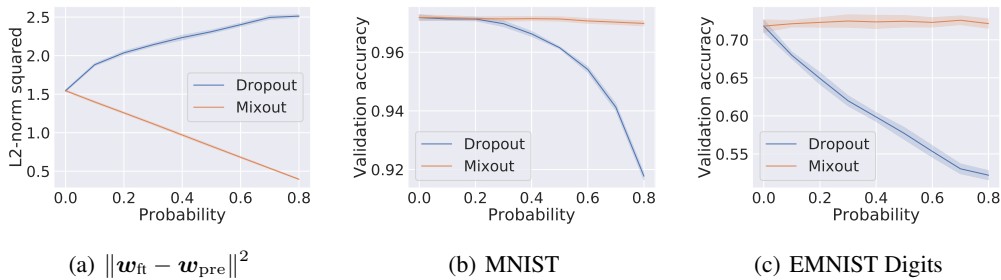

(a) $\|\boldsymbol{w}_{\text{ft}} - \boldsymbol{w}_{\text{pre}}\|^2$        (b) MNIST        (c) EMNIST Digits

Figure 2: We present $\|\boldsymbol{w}_{\text{ft}} - \boldsymbol{w}_{\text{pre}}\|^2$, validation accuracy on MNIST (target task), and validation accuracy on EMNIST Digits (source task), as the function of the probability $p$ where $\boldsymbol{w}_{\text{ft}}$ and $\boldsymbol{w}_{\text{pre}}$ are the model parameter after finetuning and the pretrained model parameter, respectively. We report mean (curve) $\pm$ std. (shaded area) across 10 random restarts. (a): $\text{mixout}(\boldsymbol{w}_{\text{pre}}, p)$ $L^2$-penalizes the deviation from $\boldsymbol{w}_{\text{pre}}$, and this penalty becomes strong as $p$ increases. However, with $\text{dropout}(p)$, $\boldsymbol{w}_{\text{ft}}$ becomes away from $\boldsymbol{w}_{\text{pre}}$ as $p$ increases. (b): After finetuning on MNIST, both $\text{mixout}(\boldsymbol{w}_{\text{pre}}, p)$ and $\text{dropout}(p)$ result in high validation accuracy on MNIST for $p \in \{0.1, 0.2, 0.3\}$. (c): Validation accuracy of $\text{dropout}(p)$ on EMNIST Digits drops more than that of $\text{mixout}(\boldsymbol{w}_{\text{pre}}, p)$ for all $p$. $\text{mixout}(\boldsymbol{w}_{\text{pre}}, p)$ minimizes the deviation from $\boldsymbol{w}_{\text{pre}}$ and memorizes the source task better than $\text{dropout}(p)$ for all $p$.

As shown in Figure 2 (a), after finetuning with $\text{mixout}(\boldsymbol{w}_{\text{pre}}, p)$, the deviation from $\boldsymbol{w}_{\text{pre}}$ is minimized in the $L^2$-sense. This result verifies Corollary 1.1. We demonstrate that the validation accuracy of $\text{mixout}(\boldsymbol{w}_{\text{pre}}, p)$ has greater robustness to the choice of $p$ than that of $\text{dropout}(p)$. In Figure 2 (b), both $\text{dropout}(p)$ and $\text{mixout}(\boldsymbol{w}_{\text{pre}}, p)$ result in high validation accuracy on the target task (MNIST) for $p \in \{0.1, 0.2, 0.3\}$, although $\text{mixout}(\boldsymbol{w}_{\text{pre}}, p)$ is much more robust with respect to the choice of the mix probability $p$. In Figure 2 (c), the validation accuracy of $\text{mixout}(\boldsymbol{w}_{\text{pre}}, p)$ on the source task (EMNIST Digits) drops from the validation accuracy of the model at $\boldsymbol{w}_{\text{pre}}$ (0.992) to approximately 0.723 regardless of $p$. On the other hand, the validation accuracy of $\text{dropout}(p)$ on the source task respectively drops by 0.041, 0.074 and 0.105 which are more than those of $\text{mixout}(\boldsymbol{w}_{\text{pre}}, p)$ for $p \in \{0.1, 0.2, 0.3\}$.

---

[1] $\boldsymbol{w}_{\text{ft}}$ is a model parameter after finetuning.

[2] Using the same pretrained model parameter $\boldsymbol{w}_{\text{pre}}$ but perform different finetuning data shuffling.

## 5 FINETUNING A PRETRAINED LANGUAGE MODEL WITH MIXOUT

In order to experimentally validate the effectiveness of mixout, we finetune $\text{BERT}_{\text{LARGE}}$ on a subset of GLUE (Wang et al., 2018) tasks (RTE, MRPC, CoLA, and STS-B) with $\texttt{mixout}(\boldsymbol{w}_{\text{pre}})$. We choose them because Phang et al. (2018) have observed that it was unstable to finetune $\text{BERT}_{\text{LARGE}}$ on these four tasks. We use the publicly available pretrained model released by Devlin et al. (2018), ported into PyTorch by HuggingFace.[3] We use the learning setup and hyperparameters recommended by Devlin et al. (2018). We use Adam with a learning rate of $2 \times 10^{-5}$, $\beta_1 = 0.9$, $\beta_2 = 0.999$, learning rate warmup over the first 10% steps of the total steps, and linear decay of the learning rate after the warmup finishes. We train with a batch size of 32 for 3 training epochs. Since the pretrained $\text{BERT}_{\text{LARGE}}$ is the sentence encoder, we have to create an additional output layer, which is not pretrained. We initialize each parameter of it with $\mathcal{N}(0, 0.02^2)$. We describe our experimental setup further in Supplement C.2.

The original regularization strategy used in Devlin et al. (2018) for finetuning $\text{BERT}_{\text{LARGE}}$ is using both $\text{dropout}(0.1)$ and $\text{wdecay}(\boldsymbol{0}, 0.01)$ for all layers except layer normalization and intermediate layers activated by GELU (Hendrycks & Gimpel, 2016). We however cannot use $\texttt{mixout}(\boldsymbol{w}_{\text{pre}})$ nor $\text{wdecay}(\boldsymbol{w}_{\text{pre}})$ for the additional output layer which was not pretrained and therefore does not have $\boldsymbol{w}_{\text{pre}}$. We do not use any regularization for the additional output layer when finetuning $\text{BERT}_{\text{LARGE}}$ with $\texttt{mixout}(\boldsymbol{w}_{\text{pre}})$ and $\text{wdecay}(\boldsymbol{w}_{\text{pre}})$. For the other layers, we replace $\text{dropout}(0.1)$ and $\text{wdecay}(\boldsymbol{0}, 0.01)$ with $\texttt{mixout}(\boldsymbol{w}_{\text{pre}})$ and $\text{wdecay}(\boldsymbol{w}_{\text{pre}})$, respectively.

Phang et al. (2018) have reported that large pretrained models (e.g., $\text{BERT}_{\text{LARGE}}$) are prone to degenerate performance when finetuned on a task with a small number of training examples, and that multiple random restarts[4] are required to obtain a usable model better than random prediction. To compare finetuning stability of the regularization techniques, we need to demonstrate the distribution of model performance. We therefore train $\text{BERT}_{\text{LARGE}}$ with each regularization strategy on each task with 20 random restarts. We validate each random restart on the dev set to observe the behaviour of the proposed mixout and finally evaluate it on the test set for generalization. We present the test score of our proposed regularization strategy on each task in Supplement C.3.

We finetune $\text{BERT}_{\text{LARGE}}$ with $\texttt{mixout}(\boldsymbol{w}_{\text{pre}}, \{0.7, 0.8, 0.9\})$ on RTE, MRPC, CoLA, and STS-B. For the baselines, we finetune $\text{BERT}_{\text{LARGE}}$ with both $\text{dropout}(0.1)$ and $\text{wdecay}(\boldsymbol{0}, 0.01)$ as well as with $\text{wdecay}(\boldsymbol{w}_{\text{pre}}, \{0.01, 0.04, 0.07, 0.10\})$. These choices are made based on the experiments in Section 6.3 and Supplement F. In Section 6.3, we observe that finetuning $\text{BERT}_{\text{LARGE}}$ with $\texttt{mixout}(\boldsymbol{w}_{\text{pre}}, p)$ on RTE is significantly more stable with $p \in \{0.7, 0.8, 0.9\}$ while finetuning with $\text{dropout}(p)$ becomes unstable as $p$ increases. In Supplement F, we demonstrate that $\text{dropout}(0.1)$ is almost optimal for all the tasks in terms of mean dev score although Devlin et al. (2018) selected it to improve the maximum dev score.

In Figure 3, we plot the distributions of the dev scores from 20 random restarts when finetuning $\text{BERT}_{\text{LARGE}}$ with various regularization strategies on each task. For conciseness, we only show four regularization strategies; Devlin et al. (2018)'s: both $\text{dropout}(0.1)$ and $\text{wdecay}(\boldsymbol{0}, 0.01)$, Wiese et al. (2017)'s: $\text{wdecay}(\boldsymbol{w}_{\text{pre}}, 0.01)$, ours: $\texttt{mixout}(\boldsymbol{w}_{\text{pre}}, 0.7)$, and ours+Wiese et al. (2017)'s: both $\texttt{mixout}(\boldsymbol{w}_{\text{pre}}, 0.7)$ and $\text{wdecay}(\boldsymbol{w}_{\text{pre}}, 0.01)$. As shown in Figure 3 (a–c), we observe many finetuning runs that fail with the chance-level accuracy when we finetune $\text{BERT}_{\text{LARGE}}$ with both $\text{dropout}(0.1)$ and $\text{wdecay}(\boldsymbol{0}, 0.01)$ on RTE, MRPC, and CoLA. We also have a bunch of degenerate model configurations when we use $\text{wdecay}(\boldsymbol{w}_{\text{pre}}, 0.01)$ without $\texttt{mixout}(\boldsymbol{w}_{\text{pre}}, 0.7)$.

Unlike existing regularization strategies, when we use $\texttt{mixout}(\boldsymbol{w}_{\text{pre}}, 0.7)$ as a regularization technique with or without $\text{wdecay}(\boldsymbol{w}_{\text{pre}}, 0.01)$ for finetuning $\text{BERT}_{\text{LARGE}}$, the number of degenerate model configurations that fail with a chance-level accuracy significantly decreases. For example, in Figure 3 (c), we have only one degenerate model configuration when finetuning $\text{BERT}_{\text{LARGE}}$ with $\texttt{mixout}(\boldsymbol{w}_{\text{pre}}, 0.7)$ on CoLA while we observe respectively seven and six degenerate models with Devlin et al. (2018)'s and Wiese et al. (2017)'s regularization strategies.

---

[3]`https : / / s3 . amazonaws . com / models . huggingface . co / bert / bert-large-uncased-pytorch_model.bin`

[4]Using the same pretrained model parameter $\boldsymbol{w}_{\text{pre}}$ but each random restart differs from the others by shuffling target data and initializing the additional output layer differently.

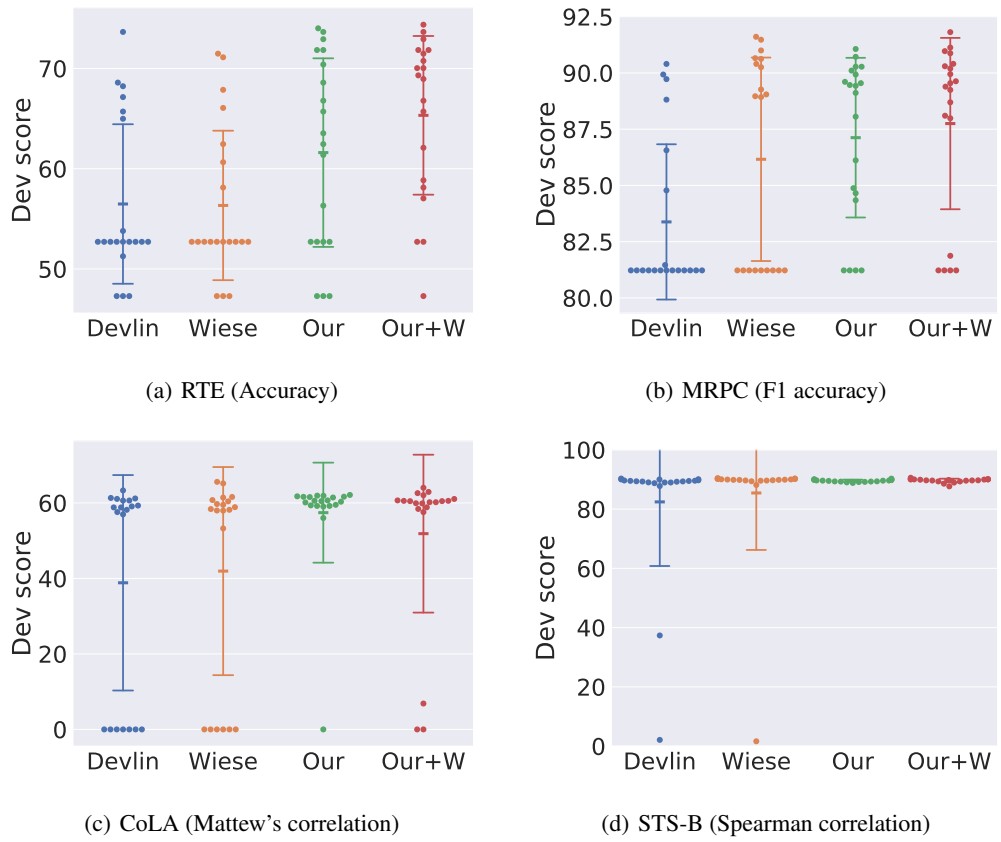

(a) RTE (Accuracy)

(b) MRPC (F1 accuracy)

(c) CoLA (Mattew's correlation)

(d) STS-B (Spearman correlation)

Figure 3: Distribution of dev scores on each task from 20 random restarts when finetuning $\text{BERT}_{\text{LARGE}}$ with Devlin et al. (2018)'s: both $\text{dropout}(0.1)$ and $\text{wdecay}(\mathbf{0}, 0.01)$, Wiese et al. (2017)'s: $\text{wdecay}(\boldsymbol{w}_{\text{pre}}, 0.01)$, ours: $\texttt{mixout}(\boldsymbol{w}_{\text{pre}}, 0.7)$, and ours+Wiese et al. (2017)'s: both $\texttt{mixout}(\boldsymbol{w}_{\text{pre}}, 0.7)$ and $\text{wdecay}(\boldsymbol{w}_{\text{pre}}, 0.01)$. We write them as Devlin (blue), Wiese (orange), Our (green), and Our+W (red), respectively. We use the same set of 20 random initializations across all the regularization setups. Error intervals show mean±std. For all the tasks, the number of finetuning runs that fail with the chance-level accuracy is significantly reduced when we use our regularization $\texttt{mixout}(\boldsymbol{w}_{\text{pre}}, 0.7)$ regardless of using $\text{wdecay}(\boldsymbol{w}_{\text{pre}}, 0.01)$.

In Figure 3 (a), we further improve the stability of finetuning $\text{BERT}_{\text{LARGE}}$ by using both $\texttt{mixout}(\boldsymbol{w}_{\text{pre}}, 0.7)$ and $\text{wdecay}(\boldsymbol{w}_{\text{pre}}, 0.01)$. Figure 3 (d) shows respectively two and one degenerate model configurations with Devlin et al. (2018)'s and Wiese et al. (2017)'s, but we do not have any degenerate resulting model with ours and ours+Wiese et al. (2017)'s. In Figure 3 (b, c), we observe that the number of degenerate model configurations increases when we use $\text{wdecay}(\boldsymbol{w}_{\text{pre}}, 0.01)$ additionally to $\texttt{mixout}(\boldsymbol{w}_{\text{pre}}, 0.7)$. In short, applying our proposed mixout significantly stabilizes the finetuning results of $\text{BERT}_{\text{LARGE}}$ on small training sets regardless of whether we use $\text{wdecay}(\boldsymbol{w}_{\text{pre}}, 0.01)$.

In Table 1, we report the average and the best dev scores across 20 random restarts for each task with various regularization strategies. The average dev scores with $\texttt{mixout}(\boldsymbol{w}_{\text{pre}}, \{0.7, 0.8, 0.9\})$ increase for all the tasks. For instance, the mean dev score of finetuning with $\texttt{mixout}(\boldsymbol{w}_{\text{pre}}, 0.8)$ on CoLA is 57.9 which is 49.2% increase over 38.8 obtained by finetuning with both $\text{dropout}(p)$ and $\text{wdecay}(\mathbf{0}, 0.01)$. We observe that using $\text{wdecay}(\boldsymbol{w}_{\text{pre}}, \{0.01, 0.04, 0.07, 0.10\})$ also improves the average dev scores for most tasks compared to using both $\text{dropout}(p)$ and $\text{wdecay}(\mathbf{0}, 0.01)$. We however observe that finetuning with $\texttt{mixout}(\boldsymbol{w}_{\text{pre}}, \{0.7, 0.8, 0.9\})$ outperforms that with $\text{wdecay}(\boldsymbol{w}_{\text{pre}}, \{0.01, 0.04, 0.07, 0.10\})$ on average. This confirms that $\texttt{mixout}(\boldsymbol{w}_{\text{pre}})$ has a different effect for finetuning $\text{BERT}_{\text{LARGE}}$ compared to $\text{wdecay}(\boldsymbol{w}_{\text{pre}})$ since $\texttt{mixout}(\boldsymbol{w}_{\text{pre}})$ is an adaptive $L^2$-regularizer along the optimization trajectory.

Since finetuning a large pretrained language model such as $\text{BERT}_{\text{LARGE}}$ on a small training set frequently fails, the final model performance has often been reported as the maximum dev score (Devlin et al., 2018; Phang et al., 2018) among a few random restarts. We thus report the best dev score for each setting in Table 1. According to the best dev scores as well, $\texttt{mixout}(\boldsymbol{w}_{\text{pre}}, \{0.7, 0.8, 0.9\})$ improves performance for all the tasks compared to using both $\text{dropout}(p)$ and $\text{wdecay}(\boldsymbol{0}, 0.01)$. For instance, using $\texttt{mixout}(\boldsymbol{w}_{\text{pre}}, 0.9)$ improves the maximum dev score by 0.9 compared to using both $\text{dropout}(p)$ and $\text{wdecay}(\boldsymbol{0}, 0.01)$ on MRPC. Unlike the average dev scores, the best dev scores achieved by using $\text{wdecay}(\boldsymbol{w}_{\text{pre}}, \{0.01, 0.04, 0.07, 0.10\})$ are better than those achieved by using $\texttt{mixout}(\boldsymbol{w}_{\text{pre}}, \{0.7, 0.8, 0.9\})$ except RTE on which it was better to use $\texttt{mixout}(\boldsymbol{w}_{\text{pre}}, \{0.7, 0.8, 0.9\})$ than $\text{wdecay}(\boldsymbol{w}_{\text{pre}}, \{0.01, 0.04, 0.07, 0.10\})$.

Table 1: Mean (*max*) dev scores across 20 random restarts when finetuning $\text{BERT}_{\text{LARGE}}$ with various regularization strategies on each task. We show the following baseline results on the first and second cells: Devlin et al. (2018)'s regularization strategy (both $\text{dropout}(p)$ and $\text{wdecay}(\boldsymbol{0}, 0.01)$) and Wiese et al. (2017)'s regularization strategy ($\text{wdecay}(\boldsymbol{w}_{\text{pre}}, \{0.01, 0.04, 0.07, 0.10\})$). In the third cell, we demonstrate finetuning results with only $\texttt{mixout}(\boldsymbol{w}_{\text{pre}}, \{0.7, 0.8, 0.9\})$. The results with both $\texttt{mixout}(\boldsymbol{w}_{\text{pre}}, \{0.7, 0.8, 0.9\})$ and $\text{wdecay}(\boldsymbol{w}_{\text{pre}}, 0.01)$ are also presented in the fourth cell. **Bold** marks the best of each statistics within each column. The mean dev scores greatly increase for all the tasks when we use $\texttt{mixout}(\boldsymbol{w}_{\text{pre}}, \{0.7, 0.8, 0.9\})$.

| TECHNIQUE 1 | TECHNIQUE 2 | RTE | MRPC | CoLA | STS-B |
|---|---|---|---|---|---|
| $\text{dropout}(0.1)$ | $\text{wdecay}(\boldsymbol{0}, 0.01)$ | 56.5 (*73.6*) | 83.4 (*90.4*) | 38.8 (*63.3*) | 82.4 (*90.3*) |
| - | $\text{wdecay}(\boldsymbol{w}_{\text{pre}}, 0.01)$ | 56.3 (*71.5*) | 86.2 (*91.6*) | 41.9 (***65.6***) | 85.4 (*90.5*) |
| - | $\text{wdecay}(\boldsymbol{w}_{\text{pre}}, 0.04)$ | 51.5 (*70.8*) | 85.8 (*91.5*) | 35.4 (*64.7*) | 80.7 (***90.6***) |
| - | $\text{wdecay}(\boldsymbol{w}_{\text{pre}}, 0.07)$ | 57.0 (*70.4*) | 85.8 (*91.0*) | 48.1 (*63.9*) | 89.6 (*90.3*) |
| - | $\text{wdecay}(\boldsymbol{w}_{\text{pre}}, 0.10)$ | 54.6 (*71.1*) | 84.2 (***91.8***) | 45.6 (*63.8*) | 84.3 (*90.1*) |
| $\texttt{mixout}(\boldsymbol{w}_{\text{pre}}, 0.7)$ | - | 61.6 (*74.0*) | 87.1 (*91.1*) | 57.4 (*62.1*) | 89.6 (*90.3*) |
| $\texttt{mixout}(\boldsymbol{w}_{\text{pre}}, 0.8)$ | - | 64.0 (*74.0*) | **89.0** (*90.7*) | 57.9 (*63.8*) | 89.4 (*90.3*) |
| $\texttt{mixout}(\boldsymbol{w}_{\text{pre}}, 0.9)$ | - | 64.3 (*73.3*) | 88.2 (*91.4*) | 55.2 (*63.4*) | 89.4 (*90.0*) |
| $\texttt{mixout}(\boldsymbol{w}_{\text{pre}}, 0.7)$ | $\text{wdecay}(\boldsymbol{w}_{\text{pre}}, 0.01)$ | **65.3** (*74.4*) | 87.8 (***91.8***) | 51.9 (*64.0*) | 89.6 (***90.6***) |
| $\texttt{mixout}(\boldsymbol{w}_{\text{pre}}, 0.8)$ | $\text{wdecay}(\boldsymbol{w}_{\text{pre}}, 0.01)$ | 62.8 (*74.0*) | 86.3 (*90.9*) | **58.3** (*65.1*) | **89.7** (*90.3*) |
| $\texttt{mixout}(\boldsymbol{w}_{\text{pre}}, 0.9)$ | $\text{wdecay}(\boldsymbol{w}_{\text{pre}}, 0.01)$ | 65.0 (***75.5***) | 88.6 (*91.3*) | 58.1 (*65.1*) | 89.5 (*90.0*) |

We investigate the effect of combining both $\texttt{mixout}(\boldsymbol{w}_{\text{pre}})$ and $\text{wdecay}(\boldsymbol{w}_{\text{pre}})$ to see whether they are complementary. We finetune $\text{BERT}_{\text{LARGE}}$ with both $\texttt{mixout}(\boldsymbol{w}_{\text{pre}}, \{0.7, 0.8, 0.9\})$ and $\text{wdecay}(\boldsymbol{w}_{\text{pre}}, 0.01)$. This leads not only to the improvement in the average dev scores but also in the best dev scores compared to using $\text{wdecay}(\boldsymbol{w}_{\text{pre}}, \{0.01, 0.04, 0.07, 0.10\})$ and using both $\text{dropout}(p)$ and $\text{wdecay}(\boldsymbol{0}, 0.01)$. The experiments in this section confirm that using $\texttt{mixout}(\boldsymbol{w}_{\text{pre}})$ as one of several regularization techniques prevents finetuning instability and yields gains in dev scores.

# 6 ABLATION STUDY

In this section, we perform ablation experiments to better understand $\texttt{mixout}(\boldsymbol{w}_{\text{pre}})$. Unless explicitly stated, all experimental setups are the same as in Section 5.

## 6.1 MIXOUT WITH A SUFFICIENT NUMBER OF TRAINING EXAMPLES

We showed the effectiveness of the proposed mixout finetuning with only a few training examples in Section 5. In this section, we investigate the effectiveness of the proposed mixout in the case of a larger finetuning set. Since it has been stable to finetune $\text{BERT}_{\text{LARGE}}$ on a sufficient number of training examples (Devlin et al., 2018; Phang et al., 2018), we expect to see the change in the behaviour of $\texttt{mixout}(\boldsymbol{w}_{\text{pre}})$ when we use it to finetune $\text{BERT}_{\text{LARGE}}$ on a larger training set.

We train $\text{BERT}_{\text{LARGE}}$ by using both $\texttt{mixout}(\boldsymbol{w}_{\text{pre}}, 0.7)$ and $\text{wdecay}(\boldsymbol{w}_{\text{pre}}, 0.01)$ with 20 random restarts on SST-2.[5] We also train $\text{BERT}_{\text{LARGE}}$ by using both $\text{dropout}(p)$ and $\text{wdecay}(\boldsymbol{0}, 0.01)$ with 20 random restarts on SST-2 as the baseline. In Table 2, we report the mean and maximum of

---

[5]For the description of SST-2, see Supplement C.2.

SST-2 dev scores across 20 random restarts with each regularization strategy. We observe that there is little difference between their mean and maximum dev scores on a larger training set, although using both $\mathrm{mixout}(\boldsymbol{w}_{\mathrm{pre}},\ 0.7)$ and $\mathrm{wdecay}(\boldsymbol{w}_{\mathrm{pre}},\ 0.01)$ outperformed using both $\mathrm{dropout}(p)$ and $\mathrm{wdecay}(\boldsymbol{0},\ 0.01)$ on the small training sets in Section 5.

Table 2: Mean (*max*) SST-2 dev scores across 20 random restarts when finetuning $\mathrm{BERT}_{\mathrm{LARGE}}$ with each regularization strategy. **Bold** marks the best of each statistics within each column. For a large training set, both mean and maximum dev scores are similar to each other.

| TECHNIQUE 1 | TECHNIQUE 2 | SST-2 |
|---|---|---|
| $\mathrm{dropout}(0.1)$ | $\mathrm{wdecay}(\boldsymbol{0},\ 0.01)$ | 93.4 (*94.0*) |
| $\mathrm{mixout}(\boldsymbol{w}_{\mathrm{pre}},\ 0.7)$ | $\mathrm{wdecay}(\boldsymbol{w}_{\mathrm{pre}},\ 0.01)$ | **93.5** (***94.3***) |

## 6.2 EFFECT OF A REGULARIZATION TECHNIQUE FOR AN ADDITIONAL OUTPUT LAYER

In this section, we explore the effect of a regularization technique for an additional output layer. There are two regularization techniques available for the additional output layer: $\mathrm{dropout}(p)$ and $\mathrm{mixout}(\boldsymbol{w}_0,\ p)$ where $\boldsymbol{w}_0$ is its randomly initialized parameter. Either of these strategies differs from the earlier experiments in Section 5 where we did not put any regularization for the additional output layer.

Table 3: We present mean (*max*) dev scores across 20 random restarts with various regularization techniques for the additional output layers (ADDITIONAL) when finetuning $\mathrm{BERT}_{\mathrm{LARGE}}$ on each task. For all cases, we apply $\mathrm{mixout}(\boldsymbol{w}_{\mathrm{pre}},\ 0.7)$ to the pretrained layers (PRETRAINED). The first row corresponds to the setup in Section 5. In the second row, we apply $\mathrm{mixout}(\boldsymbol{w}_0,\ 0.7)$ to the additional output layer where $\boldsymbol{w}_0$ is its randomly initialized parameter. The third row shows the results obtained by applying $\mathrm{dropout}(0.7)$ to the additional output layer. In the fourth row, we demonstrate the best of each result from all the regularization strategies shown in Table 1. **Bold** marks the best of each statistics within each column. We obtain additional gains in dev scores by varying the regularization technique for the additional output layer.

| PRETRAINED | ADDITIONAL | RTE | MRPC | CoLA | STS-B |
|---|---|---|---|---|---|
| $\mathrm{mixout}(\boldsymbol{w}_{\mathrm{pre}},\ 0.7)$ | - | 61.6 (*74.0*) | 87.1 (*91.1*) | 57.4 (*62.1*) | 89.6 (*90.3*) |
| $\mathrm{mixout}(\boldsymbol{w}_{\mathrm{pre}},\ 0.7)$ | $\mathrm{mixout}(\boldsymbol{w}_0,\ 0.7)$ | **66.5** (***75.5***) | 88.1 (*92.4*) | **58.7** (***65.6***) | **89.7** (***90.6***) |
| $\mathrm{mixout}(\boldsymbol{w}_{\mathrm{pre}},\ 0.7)$ | $\mathrm{dropout}(0.7)$ | 57.2 (*70.8*) | 85.9 (***92.5***) | 48.9 (*64.3*) | 89.2 (*89.8*) |
| The best of each result from Table 1 | | 65.3 (***75.5***) | **89.0** (*91.8*) | 58.3 (***65.6***) | **89.7** (***90.6***) |

We report the average and best dev scores across 20 random restarts when finetuning $\mathrm{BERT}_{\mathrm{LARGE}}$ with $\mathrm{mixout}(\boldsymbol{w}_{\mathrm{pre}},\ 0.7)$ while varying the regularization technique for the additional output layer in Table 3.[6] We observe that using $\mathrm{mixout}(\boldsymbol{w}_0,\ 0.7)$ for the additional output layer improves both the average and best dev score on RTE, CoLA, and STS-B. In the case of MRPC, we have the highest best-dev score by using $\mathrm{dropout}(0.7)$ for the additional output layer while the highest mean dev score is obtained by using $\mathrm{mixout}(\boldsymbol{w}_0,\ 0.7)$ for it. In Section 3.2, we discussed how $\mathrm{mixout}(\boldsymbol{w}_0)$ does not differ from dropout when the layer is randomly initialized, since we sample $\boldsymbol{w}_0$ from $\mathbf{w}$ whose mean and variance are $\boldsymbol{0}$ and small, respectively. Although the additional output layer is randomly initialized, we observe the significant difference between dropout and $\mathrm{mixout}(\boldsymbol{w}_0)$ in this layer. We conjecture that $\|\boldsymbol{w}_0 - \boldsymbol{0}\|$ is not sufficiently small because $\mathbb{E}\|\mathbf{w} - \boldsymbol{0}\|$ is proportional to the dimensionality of the layer (2,048). We therefore expect $\mathrm{mixout}(\boldsymbol{w}_0)$ to behave differently from dropout even for the case of training from scratch.

In the last row of Table 3, we present the best of the corresponding result from Table 1. We have the highest mean and best dev scores when we respectively use $\mathrm{mixout}(\boldsymbol{w}_{\mathrm{pre}},\ 0.7)$ and $\mathrm{mixout}(\boldsymbol{w}_0,\ 0.7)$ for the pretrained layers and the additional output layer on RTE, CoLA, and STS-B. The highest mean dev score on MRPC is obtained by using $\mathrm{mixout}(\boldsymbol{w}_{\mathrm{pre}},\ 0.8)$ for the pretrained layers which is one of the results in Table 1. We have the highest best dev score on MRPC when we use $\mathrm{mixout}(\boldsymbol{w}_{\mathrm{pre}},\ 0.7)$ and $\mathrm{dropout}(0.7)$ for the pretrained layers and the additional output

---

[6]In this experiment, we use neither $\mathrm{wdecay}(\boldsymbol{0})$ nor $\mathrm{wdecay}(\boldsymbol{w}_{\mathrm{pre}})$.

layer, respectively. The experiments in this section reveal that using $\texttt{mixout}(\boldsymbol{w}_0)$ for a randomly initialized layer of a pretrained model is one of the regularization schemes to improve the average dev score and the best dev score.

### 6.3 EFFECT OF MIX PROBABILITY FOR MIXOUT AND DROPOUT

We explore the effect of the hyperparameter $p$ when finetuning $\text{BERT}_{\text{LARGE}}$ with $\texttt{mixout}(\boldsymbol{w}_{\text{pre}}, p)$ and $\text{dropout}(p)$. We train $\text{BERT}_{\text{LARGE}}$ with $\texttt{mixout}(\boldsymbol{w}_{\text{pre}}, \{0.0, 0.1, \cdots, 0.9\})$ on RTE with 20 random restarts. We also train $\text{BERT}_{\text{LARGE}}$ after replacing $\texttt{mixout}(\boldsymbol{w}_{\text{pre}}, p)$ by $\text{dropout}(p)$ with 20 random restarts. We do not use any regularization technique for the additional output layer. Because we use neither $\text{wdecay}(\boldsymbol{0})$ nor $\text{wdecay}(\boldsymbol{w}_{\text{pre}})$ in this section, $\text{dropout}(0.0)$ and $\texttt{mixout}(\boldsymbol{w}_{\text{pre}}, 0.0)$ are equivalent to finetuning without regularization.

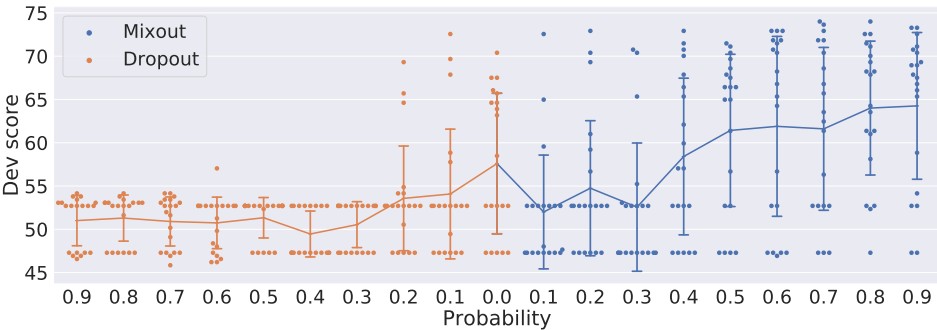

Figure 4: Distribution of RTE dev scores (Accuracy) from 20 random restarts when finetuning $\text{BERT}_{\text{LARGE}}$ with $\text{dropout}(p)$ (orange) or $\texttt{mixout}(\boldsymbol{w}_{\text{pre}}, p)$ (blue). Error intervals show mean±std. We do not use $\text{wdecay}(\boldsymbol{0})$ nor $\text{wdecay}(\boldsymbol{w}_{\text{pre}})$. In the case of $\texttt{mixout}(\boldsymbol{w}_{\text{pre}}, p)$, the number of usable models after finetuning with $\texttt{mixout}(\boldsymbol{w}_{\text{pre}}, \{0.7, 0.8, 0.9\})$ is significantly more than the number of usable models after finetuning with $\text{dropout}(p)$ for all $p$.

It is not helpful to vary $p$ for $\text{dropout}(p)$ while $\texttt{mixout}(\boldsymbol{w}_{\text{pre}}, p)$ helps significantly in a wide range of $p$. Figure 4 shows distributions of RTE dev scores across 20 random restarts when finetuning $\text{BERT}_{\text{LARGE}}$ with $\text{dropout}(p)$ and $\texttt{mixout}(\boldsymbol{w}_{\text{pre}}, p)$ for $p \in \{0.0, 0.1, \cdots, 0.9\}$. The mean dev score of finetuning $\text{BERT}_{\text{LARGE}}$ with $\texttt{mixout}(\boldsymbol{w}_{\text{pre}}, p)$ increases as $p$ increases. On the other hand, the mean dev score of finetuning $\text{BERT}_{\text{LARGE}}$ with $\text{dropout}(p)$ decreases as $p$ increases. If $p$ is less than 0.4, finetuning with $\texttt{mixout}(\boldsymbol{w}_{\text{pre}}, p)$ does not improve the finetuning results of using $\text{dropout}(\{0.0, 0.1, 0.2\})$. We however observe that $\texttt{mixout}(\boldsymbol{w}_{\text{pre}}, \{0.7, 0.8, 0.9\})$ yields better average dev scores than $\text{dropout}(p)$ for all $p$, and significantly reduces the number of finetuning runs that fail with the chance-level accuracy.

We notice that the proposed mixout spends more time than dropout from the experiments in this section. It takes longer to finetune a model with the proposed mixout than with the original dropout, although this increase is not significant especially considering the waste of time from failed finetuning runs using dropout. In Supplement E, we describe more in detail the difference between mixout and dropout in terms of wall-clock time.

## 7 CONCLUSION

The special case of our approach, $\texttt{mixout}(\boldsymbol{w}_{\text{pre}})$, is one of several regularization techniques modifying a finetuning procedure to prevent catastrophic forgetting. Unlike $\text{wdecay}(\boldsymbol{w}_{\text{pre}})$ proposed earlier by Wiese et al. (2017), $\texttt{mixout}(\boldsymbol{w}_{\text{pre}})$ is an adaptive $L^2$-regularizer toward $\boldsymbol{w}_{\text{pre}}$ in the sense that its regularization coefficient adapts along the optimization path. Due to this difference, the proposed mixout improves the stability of finetuning a big, pretrained language model even with only a few training examples of a target task. Furthermore, our experiments have revealed the proposed approach improves finetuning results in terms of the average accuracy and the best accuracy over multiple runs. We emphasize that our approach can be used with any pretrained language models such as RoBERTa (Liu et al., 2019) and XLNet (Yang et al., 2019), since mixout does not depend on model architectures, and leave it as future work.

ACKNOWLEDGMENTS

The first and third authors' work was supported by the National Research Foundation of Korea (NRF) grants funded by the Korea government (MOE, MSIT) (NRF-2017R1A2B4011546, NRF-2019R1A5A1028324). The second author thanks support by AdeptMind, eBay, TenCent, NVIDIA and CIFAR and was partly supported by Samsung Electronics (Improving Deep Learning using Latent Structure).

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

SUPPLEMENTARY MATERIAL

## A    PROOFS FOR THEOREM 1

**Theorem 1.** *Assume that the loss function $\mathcal{L}$ is strongly convex. Suppose that a random mixture function with respect to $w$ from $u$ and $M$ is given by*

$$\Phi(\boldsymbol{w};\, \boldsymbol{u}, \boldsymbol{M}) = \mu^{-1}\big((\boldsymbol{I} - \boldsymbol{M})\boldsymbol{u} + \boldsymbol{M}\boldsymbol{w} - (1-\mu)\boldsymbol{u}\big),$$

*where $\boldsymbol{M}$ is $\mathrm{diag}(M_1,\, M_2,\, \cdots,\, M_d)$ satisfying $\mathbb{E}M_i = \mu$ and $\mathrm{Var}(M_i) = \sigma^2$ for all $i$. Then, there exists $m > 0$ such that*

$$\mathbb{E}\mathcal{L}\big(\Phi(\boldsymbol{w};\, \boldsymbol{u}, \boldsymbol{M})\big) \geq \mathcal{L}(\boldsymbol{w}) + \frac{m\sigma^2}{2\mu^2}\|\boldsymbol{w} - \boldsymbol{u}\|^2, \tag{11}$$

*for all $\boldsymbol{w}$.*

*Proof.* Since $\mathcal{L}$ is strongly convex, there exist $m > 0$ such that

$$\mathbb{E}\mathcal{L}\big(\Phi(\boldsymbol{w};\, \boldsymbol{u}, \boldsymbol{M})\big) = \mathbb{E}\mathcal{L}\Big(\boldsymbol{w} + \big(\Phi(\boldsymbol{w};\, \boldsymbol{u}, \boldsymbol{M}) - \boldsymbol{w}\big)\Big)$$

$$\geq \mathcal{L}(\boldsymbol{w}) + \nabla\mathcal{L}(\boldsymbol{w})^\top \mathbb{E}[\Phi(\boldsymbol{w};\, \boldsymbol{u}, \boldsymbol{M}) - \boldsymbol{w}] + \frac{m}{2}\mathbb{E}\|\Phi(\boldsymbol{w};\, \boldsymbol{u}, \boldsymbol{M}) - \boldsymbol{w}\|^2, \tag{12}$$

for all $\boldsymbol{w}$ by equation 1. Recall that $\mathbb{E}M_i = \mu$ and $\mathrm{Var}(M_i) = \sigma^2$ for all $i$. Then, we have

$$\mathbb{E}[\Phi(\boldsymbol{w};\, \boldsymbol{u}, \boldsymbol{M}) - \boldsymbol{w}] = \boldsymbol{0}, \tag{13}$$

and

$$\mathbb{E}\|\Phi(\boldsymbol{w};\, \boldsymbol{u}, \boldsymbol{M}) - \boldsymbol{w}\|^2 = \mathbb{E}\left\|\frac{1}{\mu}(\boldsymbol{w} - \boldsymbol{u})(\boldsymbol{M} - \mu\boldsymbol{I})\right\|^2$$

$$= \frac{1}{\mu^2}\sum_{i=1}^d (w_i - u_i)^2 \mathbb{E}(M_i - \mu)^2$$

$$= \frac{\sigma^2}{\mu^2}\|\boldsymbol{w} - \boldsymbol{u}\|^2. \tag{14}$$

By using equation 13 and equation 14, we can rewrite equation 12 as

$$\mathbb{E}\mathcal{L}\big(\Phi(\boldsymbol{w};\, \boldsymbol{u}, \boldsymbol{M})\big) \geq \mathcal{L}(\boldsymbol{w}) + \frac{m\sigma^2}{2\mu^2}\|\boldsymbol{w} - \boldsymbol{u}\|^2.$$

$\square$

## B    APPLYING TO SPECIFIC LAYERS

We often apply dropout to specific layers. For instance, Simonyan & Zisserman (2014) applied dropout to fully connected layers only. We generalize Theorem 1 to the case in which mixconnect is only applied to specific layers, and it can be done by constructing $\boldsymbol{M}$ in a particular way. To better characterize mixconnect applied to specific layers, we define the index set $\mathbb{I}$ as $\mathbb{I} = \{i :\, M_i = 1\}$. Furthermore, we use $\tilde{\boldsymbol{w}}$ and $\tilde{\boldsymbol{u}}$ to denote $(w_i)_{i\notin\mathbb{I}}$ and $(u_i)_{i\notin\mathbb{I}}$, respectively. Then, we generalize equation 7 to

$$\mathbb{E}\mathcal{L}\big(\Phi(\boldsymbol{w};\, \boldsymbol{u}, \boldsymbol{M})\big) \geq \mathcal{L}(\boldsymbol{w}) + \frac{m\sigma^2}{2\mu^2}\|\tilde{\boldsymbol{w}} - \tilde{\boldsymbol{u}}\|^2. \tag{15}$$

From equation 15, applying $\mathtt{mixconnect}(\boldsymbol{u},\, \mu,\, \sigma^2)$ is to use adaptive $\mathrm{wdecay}(\tilde{\boldsymbol{u}})$ on the weight parameter of the specific layers $\tilde{\boldsymbol{w}}$. Similarly, we can regard applying $\mathtt{mixout}(\boldsymbol{u},\, p)$ to specific layers as adaptive $\mathrm{wdecay}(\tilde{\boldsymbol{u}})$.

## C    EXPERIMENTAL DETAILS

### C.1    FROM EMNIST DIGITS TO MNIST

**Model Architecture**    The model architecture in Section 4 is a 784-300-100-10 fully connected network with a softmax output layer. For each hidden layer, we add layer normalization (Ba et al., 2016) right after the ReLU (Nair & Hinton, 2010) nonlinearity. We initialize each parameter with $\mathcal{N}(0, 0.02^2)$ and each bias with 0.

**Regularization**    In the pretraining stage, we use $\mathrm{dropout}(0.1)$ and $\mathrm{wdecay}(\mathbf{0}, 0.01)$. We apply $\mathrm{dropout}(0.1)$ to all hidden layers. That is, we do not drop neurons of the input and output layers. $\mathrm{wdecay}(\mathbf{0}, 0.01)$ does not penalize the parameters for bias and layer normalization. When we finetune our model on MNIST, we replace $\mathrm{dropout}(p)$ with $\mathtt{mixout}(\boldsymbol{w}_{\mathrm{pre}}, p)$. We use neither $\mathrm{wdecay}(\mathbf{0})$ nor $\mathrm{wdecay}(\boldsymbol{w}_{\mathrm{pre}})$ for finetuning.

**Dataset**    For pretraining, we train our model on EMNIST Digits. This dataset has 280,000 characters into 10 balanced classes. These characters are compatible with MNIST characters. EMNIST Digits provides 240,000 characters for training and 40,000 characters for test. We use 240,000 characters provide for training and split these into the training set (216,000 characters) and validation set (24,000 characters). For finetuning, we train our model on MNIST. This has 70,000 characters into 10 balance classes. MNIST provide 60,000 characters for training and 10,000 characters for test. We use 60,000 characters given for training and split these into the training set (54,000 characters) and validation set (6,000 characters).

**Data Preprocessing**    We only use normalization after scaling pixel values into $[0, 1]$. We do not use any data augmentation.

### C.2    FINETUNING BERT ON PARTIAL GLUE TASKS

**Model Architecture**    Because the model architecture of $\mathrm{BERT}_{\mathrm{LARGE}}$ is identical to the original (Devlin et al., 2018), we omit its exhaustive description. Briefly, $\mathrm{BERT}_{\mathrm{LARGE}}$ has 24 layers, 1024 hidden size, and 16 self-attention heads (total 340M parameters). We use the publicly available pretrained model released by Devlin et al. (2018), ported into PyTorch by HuggingFace.[7] We initialize each weight parameter and bias for an additional output layer with $\mathcal{N}(0, 0.02^2)$ and 0, respectively.

**Regularization**    In the finetuning stage, Devlin et al. (2018) used $\mathrm{wdecay}(\mathbf{0}, 0.01)$ for all parameters except bias and layer normalization. They apply $\mathrm{dropout}(0.1)$ to all layers except each hidden layer activated by GELU (Hendrycks & Gimpel, 2016) and layer normalization. We substitute $\mathrm{wdecay}(\boldsymbol{w}_{\mathrm{pre}})$ and $\mathtt{mixout}(\boldsymbol{w}_{\mathrm{pre}})$ for $\mathrm{wdecay}(\mathbf{0}, 0.01)$ and $\mathrm{dropout}(0.1)$, respectively.

**Dataset**    We use a subset of GLUE (Wang et al., 2018) tasks. The brief description for each dataset is as the following:

- **RTE** (2,500 training examples): Binary entailment task (Dagan et al., 2006)
- **MRPC** (3,700 training examples): Semantic similarity (Dolan & Brockett, 2005)
- **CoLA** (8,500 training examples): Acceptability classification (Warstadt et al., 2018)
- **STS-B** (7,000 training examples): Semantic textual similarity (Cer et al., 2017)
- **SST-2** (67,000 training examples): Binary sentiment classification (Socher et al., 2013)

In this paper, we reported F1 accuracy scores for MRPC, Mattew's correlation scores for CoLA, Spearman correlation scores for STS-B, and accuracy scores for the other tasks.

**Data Preprocessing**    We use the publicly available implementation of $\mathrm{BERT}_{\mathrm{LARGE}}$ by Hugging-Face.[8]

---

[7]`https : / / s3 . amazonaws . com / models . huggingface . co / bert / bert-large-uncased-pytorch_model.bin`

[8]`https://github.com/huggingface/pytorch-transformers`

## C.3 TEST RESULTS ON GLUE TASKS

We expect that using mixout stabilizes finetuning results of $\text{BERT}_{\text{LARGE}}$ on a small training set. To show this, we demonstrated distributions of dev scores from 20 random restarts on RTE, MRPC, CoLA, and STS-B in Figure 3. We further obtained the highest average/best dev score on each task in Table 3. To confirm the generalization of the our best model on the dev set, we demonstrate the test results scored by the evaluation server[9] in Table 4.

Table 4: We present the test score when finetuning $\text{BERT}_{\text{LARGE}}$ with each regularization strategy on each task. The first row shows the test scores obtained by using both $\text{dropout}(p)$ and $\text{wdecay}(\mathbf{0},\ 0.01)$. These results in the first row are reported by Devlin et al. (2018). They used the learning rate of $\{2 \times 10^{-5},\ 3 \times 10^{-5},\ 4 \times 10^{-5},\ 5 \times 10^{-5}\}$ and a batch size of 32 for 3 epochs with multiple random restarts. They selected the best model on each dev set. In the second row, we demonstrate the test scores obtained by using the proposed mixout in Section 6.2: using $\text{mixout}(\mathbf{w}_{\text{pre}},\ 0.7)$ for the pretrained layers and $\text{mixout}(\mathbf{w}_0,\ 0.7)$ for the additional output layer where $\mathbf{w}_0$ is its randomly initialized weight parameter. We used the learning rate of $2 \times 10^{-5}$ and a batch size of 32 for 3 epochs with 20 random restarts. We submitted the best model on each dev set. The third row shows that the test scores obtained by using both $\text{dropout}(p)$ and $\text{wdecay}(\mathbf{0},\ 0.01)$ with same experimental setups of the second row. **Bold** marks the best within each column. The proposed mixout improves the test scores except MRPC compared to the original regularization strategy proposed by Devlin et al. (2018).

| STRATEGY | RTE | MRPC | CoLA | STS-B |
|:---:|:---:|:---:|:---:|:---:|
| Devlin et al. (2018) | 70.1 | **89.3** | 60.5 | 86.5 |
| $\text{mixout}(\mathbf{w}_{\text{pre}},\ 0.7)$ & $\text{mixout}(\mathbf{w}_0,\ 0.7)$ | **70.2** | 89.1 | **62.1** | **87.3** |
| $\text{dropout}(p) + \text{wdecay}(\mathbf{0},\ 0.01)$ | 68.2 | 88.3 | 59.6 | 86.0 |

For all the tasks except MRPC, the test scores obtained by the proposed mixout[10] are better than those reported by Devlin et al. (2018). We explored the behaviour of finetuning $\text{BERT}_{\text{LARGE}}$ with mixout by using the learning rate of $2 \times 10^{-5}$ while Devlin et al. (2018) obtained their results by using the learning rate of $\{2 \times 10^{-5},\ 3 \times 10^{-5},\ 4 \times 10^{-5},\ 5 \times 10^{-5}\}$. We thus present the test scores obtained by the regularization strategy of Devlin et al. (2018) when the learning rate is $2 \times 10^{-5}$. The results in this section show that the best model on the dev set generalizes well, and all the experiments based on dev scores in this paper are proper to validate the effectiveness of the proposed mixout. For the remaining GLUE tasks such as SST-2 with a sufficient number of training instances, we observed that using mixout does not differs from using dropout in Section 6.1. We therefore omit the test results on the other tasks in GLUE.

## D VERIFICATION OF COROLLARY 1.1 WITH LEAST SQUARES REGRESSION

Corollary 1.1 shows that $\text{mixout}(\mathbf{u},\ p)$ regularizes learning to minimize the deviation from the target model parameter $\mathbf{u}$, and the strength of regularization increases as $p$ increases when the loss function is strongly convex. In order to validate this, we explore the behavior of least squares regression with $\text{mixout}(\mathbf{u},\ p)$ on a synthetic dataset. For randomly given $w_1^*$ and $w_2^*$, we generated an observation $y$ satisfying $y = w_1^* x + w_2^* + \epsilon$ where $\epsilon$ is Gaussian noise. We set the model to $\hat{y} = w_1 x + w_2$. That is, the model parameter $\mathbf{w}$ is given by $(w_1,\ w_2)$. We randomly pick $\mathbf{u}$ as a target model parameter for $\text{mixout}(\mathbf{u},\ p)$ and perform least squares regression with $\text{mixout}(\mathbf{u},\ \{0.0,\ 0.3,\ 0.6,\ 0.9\})$. As shown in Figure 5, $\mathbf{w}$ converges to the target model parameter $\mathbf{u}$ rather than the true model parameter $\mathbf{w}^* = (w_1^*,\ w_2^*)$ as the mix probability $p$ increases.

---

[9] https://gluebenchmark.com/leaderboard

[10] The regularization strategy in Section 6.2: using $\text{mixout}(\mathbf{w}_{\text{pre}},\ 0.7)$ for the pretrained layers and $\text{mixout}(\mathbf{w}_0,\ 0.7)$ for the additional output where $\mathbf{w}_0$ is its randomly initialized weight parameter.

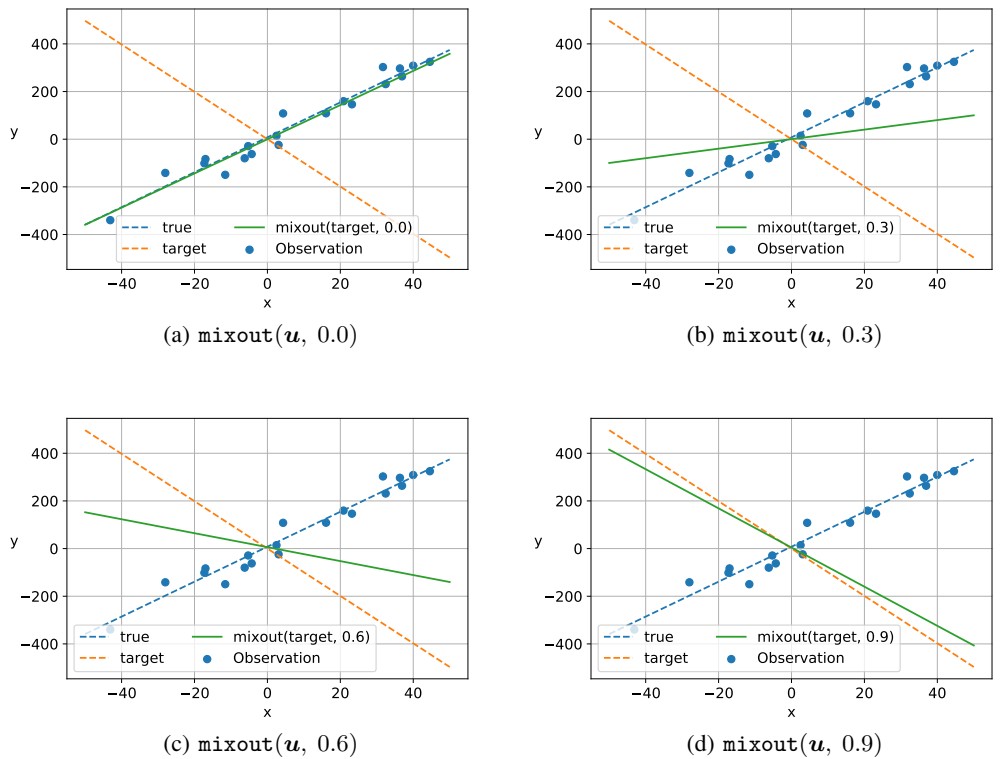

(a) mixout($\boldsymbol{u}$, 0.0)     (b) mixout($\boldsymbol{u}$, 0.3)

(c) mixout($\boldsymbol{u}$, 0.6)     (d) mixout($\boldsymbol{u}$, 0.9)

Figure 5: Behavior of mixout($\boldsymbol{u}$, $p$) for a strongly convex loss function. We plot the line obtained by least squares regression with mixout($\boldsymbol{u}$, {0.0, 0.3, 0.6, 0.9}) (each green line) on a synthetic dataset (blue dots) generated by the true line (each blue dotted line). As $p$ increases, the regression line (each green line) converges to the target line generated by the target model parameter $\boldsymbol{u}$ (each orange dotted line) rather than the true line (each blue dotted line).

## E    TIME USAGE OF MIXOUT COMPARED TO DROPOUT

We recorded the training time of the experiment in Section 6.3 to compare the time usage of mixout and that of dropout. It took about 843 seconds to finetune $\text{BERT}_{\text{LARGE}}$ with mixout($\boldsymbol{w}_{\text{pre}}$). On the other hand, it took about 636 seconds to finetune $\text{BERT}_{\text{LARGE}}$ with dropout. mixout($\boldsymbol{w}_{\text{pre}}$) spends 32.5% more time than dropout since mixout($\boldsymbol{w}_{\text{pre}}$) needs an additional computation with the pretrained model parameter $\boldsymbol{w}_{\text{pre}}$. However, as shown in Figure 4, at least 15 finetuning runs among 20 random restarts fail with the chance-level accuracy on RTE with dropout($p$) for all $p$ while only 4 finetuning runs out of 20 random restarts are unusable with mixout($\boldsymbol{w}_{\text{pre}}$, 0.8). From this result, it is reasonable to finetune with the proposed mixout although this requires additional time usage compared to dropout.

## F    EXTENSIVE HYPERPARAMETER SEARCH FOR DROPOUT

Devlin et al. (2018) finetuned $\text{BERT}_{\text{LARGE}}$ with dropout(0.1) on all GLUE (Wang et al., 2018) tasks. They chose it to improve the maximum dev score on each downstream task, but we have reported not only the maximum dev score but also the mean dev score to quantitatively compare various regularization techniques in our paper. In this section, we explore the effect of the hyper-parameter $p$ when finetuning $\text{BERT}_{\text{LARGE}}$ with dropout($p$) on RTE, MRPC, CoLA, and STS-B to show dropout(0.1) is optimal in terms of mean dev score. All experimental setups for these experiments are the same as Section 6.3.

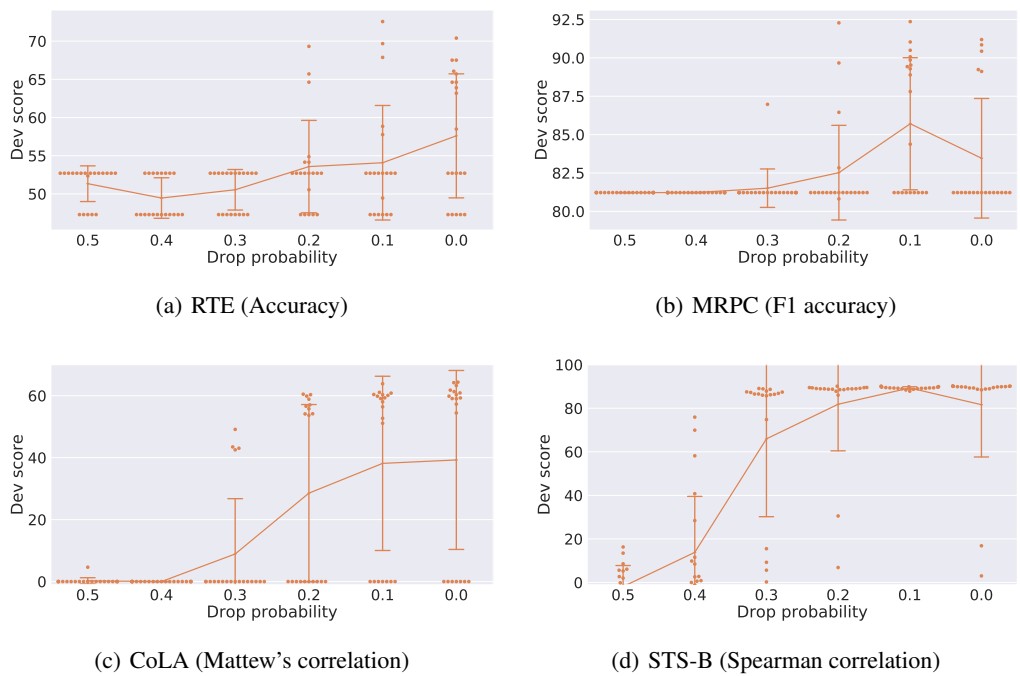

(a) RTE (Accuracy)      (b) MRPC (F1 accuracy)

(c) CoLA (Mattew's correlation)    (d) STS-B (Spearman correlation)

Figure 6: Distribution of dev scores on each task from 20 random restarts when finetuning BERT$_{\mathrm{LARGE}}$ with dropout($\{0.0, 0.1, \cdots, 0.5\}$). Error intervals show mean$\pm$std. When we use dropout(0.1), we have the highest average dev scores on MRPC and STS-B and the second-highest average dev scores on RTE and CoLA. These results show that dropout(0.1) is almost optimal for all tasks in terms of mean dev score.

As shown in Figure 6, we have the highest average dev score on MRPC with dropout(0.1) as well as on STS-B. We obtain the highest average dev scores with dropout(0.0) on RTE and CoLA, but we get the second-highest average dev scores with dropout(0.1) on them. These experiments confirm that the drop probability 0.1 is almost optimal for the highest average dev score on each task.

