# OpenReview forum: "Mixout: Effective Regularization to Finetune Large-scale Pretrained Language Models"
_ICLR.cc/2020/Conference — Accept (Poster)_

### Official Review · AnonReviewer1 · 2019-10-23
**Official Blind Review #1**

**Rating:** 6

**Review:**

This paper introduces a new regularization technique refered as “mixout”, motivated by dropout. Mixout stochastically mixes the parameters of two models. Experiments shows the stability of finetuning and the method greatly improve the average accuracy.

I really like the proposed idea, and the paper is easy to understand and follow, and the experiments are well designed.

The time usage of the regularization is not discussed. It seems the method needs to maintain two copies of parameters, it would be much better if the author can provide the time usage of the experiments.

**Experience Assessment:**

I have read many papers in this area.

**Review Assessment: Checking Correctness Of Derivations And Theory:**

I did not assess the derivations or theory.

**Review Assessment: Checking Correctness Of Experiments:**

I assessed the sensibility of the experiments.

**Review Assessment: Thoroughness In Paper Reading:**

I read the paper at least twice and used my best judgement in assessing the paper.

---

> ### Author Response · Authors · 2019-11-11
> **Response to AnonReviewer1**
>
> Thank you for your comments!
>
> “The time usage of the regularization is not discussed. It seems the method needs to maintain two copies of parameters, it would be much better if the author can provide the time usage of the experiments.”
>
> We strongly agree with you. The time usage discussion of the proposed mixout is useful information for the readers. We re-performed the experiments in Section 6.3  to compare the time usage of dropout and that of the proposed mixout. It took about 843 seconds to finetune BERT_LARGE with the proposed mixout. On the other hand, it took about 636 seconds to finetune BERT_LARGE with dropout. As you anticipated, the proposed mixout spent 32.5% more time than dropout. However, we emphasize at least 15 finetuning runs among 20 random restarts have chance-level accuracies on RTE with dropout(p) for all p (blue in Figure 4) while only 4 finetuning runs out of 20 random restarts are unusable with mixout(w_pre, 0.8). This fact gives a good reason to finetune with the proposed mixout although it yields the additional time usage compared to dropout.

---

### Official Review · AnonReviewer2 · 2019-11-04
**Official Blind Review #2**

**Rating:** 8

**Review:**

This paper introduces a new regularization technique “mixout” for fine-tuning BERT. Mixout technique mixes the parameters of two models — the pretrained model and the dropout model. Because it keeps pretrained model parameters in consideration all the time, it effectively prevent catastrophic forgetting. Empirical results show that mixout can stabilize fine-tuning BERT on tasks with small training examples (which has been shown to be difficult).

I’d like to accept this paper based on the extensive and detailed experiments and promising results. For example, the theory has been supported by experiment findings on handwriting dataset (computer vision) while the main contribution is on natural language tasks. The authors not only conducted experiments on tasks with less examples (the main focus of this paper) but also on a task with sufficient training examples. For each model regularization configuration, 20 random starts are used to report mean and best performance. This not only makes the results more reliable but also provides deeper insights.

Minor clarification questions:

For 20 random restarts, are they the same across regularization setups, i.e., for each dot in Figure 3, is there an orange dot that has the same initialization?

What will the extreme case, mixout(w_pre, 1.0), behave? According to Figure 1, it would always use w_pre and end up not learning on a target task. If so, would it introduce a cliff in Fig 4?


**Experience Assessment:**

I have read many papers in this area.

**Review Assessment: Checking Correctness Of Derivations And Theory:**

I did not assess the derivations or theory.

**Review Assessment: Checking Correctness Of Experiments:**

I carefully checked the experiments.

**Review Assessment: Thoroughness In Paper Reading:**

I read the paper at least twice and used my best judgement in assessing the paper.

---

> ### Author Response · Authors · 2019-11-11
> **Response to AnonReviewer2**
>
> Thank you for your comments!
>
> “For 20 random restarts, are they the same across regularization setups, i.e., for each dot in Figure 3, is there an orange dot that has the same initialization?”
>
> Yes, we use the same initialization across regularization setups. That is, the initialization of the given blue dot corresponds to each dot colored differently (orange, green, and red).
>
> “What will the extreme case, mixout(w_pre, 1.0), behave? According to Figure 1, it would always use w_pre and end up not learning on a target task. If so, would it introduce a cliff in Fig 4?”
>
> Yes, it introduces a cliff in Figure 4 when we use mixout(w_pre, 1.0). For Figure 4, we used mixout(w_pre, p) for all layers except the additional output layer because the additional output layer was not pretrained and therefore did not have w_pre. So the trainable parameters when we use mixout(w_pre, 1.0) are only the parameters of the additional output layer. We performed the experiment for this extreme case and present the distribution of dev scores from 20 random restarts in https://ibb.co/v3s0c00 . As shown in this link, the dev scores obtained by mixout(w_pre, 1.0) drops significantly. We omitted such extreme cases (dropout(1.0) and mixout(w_pre, 1.0)) to emphasize the monotonic trend of the mix/drop probability in our manuscript.

---

### Official Review · AnonReviewer4 · 2019-11-04
**Official Blind Review #4**

**Rating:** 6

**Review:**

The authors introduce a new regularization technique for the specific task of finetuning models. It's inspired by dropout and stochastically mixes source and target weights in order to avoid moving the parameters towards 0.

The authors provide a theoretical justification as to why mixout would do useful things in the convex case and then demonstrate empirically that using it achieves good accuracies on some downstream, finetuned, non-convex-loss-utilizing tasks. Their experiments incorporate both small models with good analysis (ie, sec 4) as well as larger, real-world models (sect 5). The paper is in general well-written.

I have a few concerns that I would like to see addressed:

1a. For starters, all the theoretical motivation describes a particular way in which mixout is supposed to aid in downstream tasks for the case of convex functions, but there are no experiments that match their assumptions and which demonstrate this is the actual behavior we see. It would be nice to see results which demonstrate the theory.

1b. In few of the prsented empirical experiments is it the case that the use of mixout by itself is useful.  Why does mixout have to be coupled with other regularization techniques? There is little analysis given here, either empirical or theoretical.

2. Why are there only 4 GLUE tasks reported? Devlin 2018 reports on all but WNLI.

3. The choice of hyperparameters for GLUE in sect 5 is a bit misleading. Devlin 2018 chose those parameters to get the maximum scores on downstream tasks; the metric they use is max score. However, the authors want to instead discuss the average score against a set of random restarts, perhaps because the max scores using their method aren't terribly different from the baselines.

Therefore, a more extensive hyperparameter sweep should have been run for the baselines: they should have been re-tuned for the average score if that is the metric the authors wish to use. Instead, the authors only used one task, RTE, to find baseline hyperparameters.

**Experience Assessment:**

I have read many papers in this area.

**Review Assessment: Checking Correctness Of Derivations And Theory:**

I assessed the sensibility of the derivations and theory.

**Review Assessment: Checking Correctness Of Experiments:**

I assessed the sensibility of the experiments.

**Review Assessment: Thoroughness In Paper Reading:**

I read the paper at least twice and used my best judgement in assessing the paper.

---

> ### Author Response · Authors · 2019-11-11
> **Response to AnonReviewer4 (Part I)**
>
> Thank you for your comments!
>
> “1a. For starters, all the theoretical motivation describes a particular way in which mixout is supposed to aid in downstream tasks for the case of convex functions, but there are no experiments that match their assumptions and which demonstrate this is the actual behavior we see. It would be nice to see results which demonstrate the theory.”
>
> Thank you for your insightful suggestion. Since the theoretical findings of mixout depend on the strong convexity of loss functions, it is nice to demonstrate the behavior of mixout for a strongly convex loss function. We performed the least squares regression with the synthetic dataset. For randomly given a and b, we generated observation y satisfying y = ax + b + e where e is Gaussian noise. We set the model to y_hat = w1 x + w2 (That is, the model parameter w is given by (w1, w2)) and train it with mixout(target, p) where ‘target’ is the target model parameter (u, 0). By Corollary 1.1, the final solution w* should converge to the target model parameter (u, 0) rather than the true model parameter (a, b) as the mix probability p increases. As shown in https://ibb.co/TvyHwP5 , mixout(target, p) indeed regularizes learning to minimize the deviation from the target model parameter (u, 0), and the strength of regularization increases as p increases. This ensures that our theoretical finding about the proposed mixout is valid for strongly convex functions. We will include this result in the next draft and thank you for bringing this to our attention.
>
> “1b. In few of the presented empirical experiments is it the case that the use of mixout by itself is useful.  Why does mixout have to be coupled with other regularization techniques? There is little analysis given here, either empirical or theoretical.”
>
> Thank you for your comment. We have been working with the reason why the proposed mixout has to be coupled with other regularization techniques to improve maximum dev scores. We hypothesize that BERT_LARGE (Total 300M parameters) is over-parametrized for a small training set (RTE, MRPC, STS-B, and CoLA), and more regularization is needed to avoid overfitting on those tasks. Hence, we add an additional regularization technique to regularize our finetuning runs. It is interesting to characterize the necessary strength of regularization for BERT_LARGE, and this would be great future work.
>
> “2. Why are there only 4 GLUE tasks reported? Devlin 2018 reports on all but WNLI.”
>
> The main purpose of our paper is to improve finetuning stability via proper regularization. We thus validated our regularization techniques on those 4 GLUE tasks where finetuning results were unstable. More specifically, Devlin et al. (2018) reported that BERT_LARGE needed several random restarts to avoid the unstable finetuning results on small training sets in GLUE. Phang et al. (2018) observed that such finetuning instability was shown when the number of training examples was less than 10,000. Since RTE, MRPC, CoLA, and STS-B have less than 10,000 training instances, we selected them to validate the proposed mixout.
>
> In Section 6.1, we performed the ablation experiment to explore the effectiveness of the proposed mixout on SST-2 which is one of the remaining GLUE tasks with sufficient training examples (greater than 10,000). As shown in Table 2,  we did not observe differences in terms of mean and maximum dev scores between the proposed mixout and Devlin et al. (2018)’s regularization technique. From this, we conclude that the dev scores obtained by the proposed mixout do not differ from those obtained by Devlin et al. (2018)’s when sufficient training instances are available. We therefore omitted finetuning BERT_LARGE on the GLUE tasks with more than 10,000 training data in Section 5. We will improve our presentation to be clearer in the next version.
>
> References
> [1] Jacob Devlin, Ming-Wei Chang, Kenton Lee, and Kristina Toutanova. Bert: Pre-training of deep bidirectional transformers for language understanding. arXiv preprint arXiv:1810.04805, 2018.
>
> [2] Jason Phang, Thibault F´evry, and Samuel R Bowman. Sentence encoders on stilts: Supplementary training on intermediate labeled-data tasks. arXiv preprint arXiv:1811.01088, 2018.

---

> > ### Author Response · Authors · 2019-11-11
> > **Response to AnonReviewer4 (Part II)**
> >
> > “3. The choice of hyperparameters for GLUE in sect 5 is a bit misleading. Devlin 2018 chose those parameters to get the maximum scores on downstream tasks; the metric they use is max score. However, the authors want to instead discuss the average score against a set of random restarts, perhaps because the max scores using their method aren't terribly different from the baselines. Therefore, a more extensive hyperparameter sweep should have been run for the baselines: they should have been re-tuned for the average score if that is the metric the authors wish to use. Instead, the authors only used one task, RTE, to find baseline hyperparameters.”
> >
> > We agree with you. It could be unfair to compare the average dev score between mixout(w_pre) and dropout since Devlin et al. (2018) selected the drop probability 0.1 to get the maximum dev score on each downstream task. We therefore explore the effect of drop probability for dropout when we finetune BERT_LARGE on MRPC, STS-B, and CoLA. All experimental setups for these experiments are the same as Section 6.3. As shown in https://ibb.co/ykrRMxY , we had the highest average dev score on MRPC/STS-B with dropout(0.1). We obtained the highest average dev score on CoLA with dropout(0.0), but we got almost the same average dev score with dropout(0.1) compared to dropout(0.0). From these results, we demonstrate that the hyperparameter for the baseline in our paper is almost optimal for the average dev score.
> >
> > References
> > [1] Jacob Devlin, Ming-Wei Chang, Kenton Lee, and Kristina Toutanova. Bert: Pre-training of deep bidirectional transformers for language understanding. arXiv preprint arXiv:1810.04805, 2018.

---

### Author Response · Authors · 2019-11-15
**Submission revised.**

Dear all Reviewers,

we thank you for your valuable reviews and comments. According to your suggestions, we have made more discussions and experiments to improve our paper. In particular, we added/changed the following contents:

1-R1) We have discussed the time usage to finetune BERT_LARGE with the proposed mixout compared to dropout on RTE (Section 6.3 colored in brown & Supplemental E).
2-R2) We have modified the caption of Figure 3 to clarify our experimental setups (Figure 3 colored in orange).
3-R4) We have performed the least squares regression to see the behavior of mixout when the loss function is strongly convex (Section 3.1 colored in olive & Supplemental D).
4-R4) We have run additional experiments to check whether dropout(0.1) is optimal in terms of mean dev score (Section 5 colored in olive & Supplemental F).

We hope our responses can help address your concerns and questions.

---

### Public Comment · ~Artem_Shelmanov1 · 2020-05-10
**BERT model versions issue (cased vs uncased)**

Dear authors,

1)	I have noticed that GLUE scores presented in Table 1 for “Devlin’s” regularization techniques {dropout(0.1) wdecay(0, 0.01)} on the dev set are substantially lower than Devlin’s results on the GLUE test set (e.g., for RTE: yours: 56.5; Devlin’s: 70.1). This might appear due to Devlin and you used different versions of the BERT model. You used “uncased” version, while Devlin seems to be using the “cased” version.
I ran 20 RTE experiments with the large model via Hugging face GLUE script (examples/text-classification/run_glue.py) and averaged the results:
For “uncased”, I receive: 0.553;
For “cased”, I receive: 0.689.
Therefore, I consider that the results presented in Table 1 are achieved using the inferior model, and the difference is large: > 13 pp.
Could you comment, why do you use such model? Are there improvements from mixout for better “cased” version?

2)	In Appendix C.3, you report GLUE results on the test set (Table 4). You achieve much better results than in Table 1. Additionally you use mixout(w0, 0.7), but the difference is substantial, and the difference is also significant compared to the results with mixout(w0, 0.7) in Table 3. The paper does not specify what model version is used for achieving GLUE scores on the test set.

Could you specify what model version was used for achieving GLUE scores on the test set?

Mixout looks very promising, however, this issue makes the reproduction difficult.

Best wishes,
Dr. Artem Shelmanov

---

> ### Author Response · Authors · 2020-05-10
> **Response to the public comment from Artem Shelmanov**
>
> Dear Artem Shelmanov,
>
> We used the "uncased" version of BERT_LARGE for both dev set and test set because Google Research had reported that the "uncased" pre-trained model is typically better than the "cased" except for some tasks (https://github.com/google-research/bert#pre-trained-models ). So I think that both Devlin et al. and we used the "uncased" model.
>
> You pointed out that Devlin's RTE score (56.5) presented in Table 1 was substantially lower than Devlin's test score (70.1) in Table 4 (or GLUE leader board). Devlin et al. (2018) chose the best model on the dev set and submitted it to the GLUE test server (A.3 in [1]). By this procedure, they got 70.1. Note that our 56.5 is the mean dev score over 20 runs, so it is reasonable to compare Devlin et al.'s 70.1 with our maximum Devlin's RTE score over those 20 runs (68.2), which is also reported in Table 1. There is still some difference between 70.1 and 68.2, but I would like to say this is acceptable since there can be some bias between the dev set and the test set.
>
> From our experiments, the degenerate mean dev score came from several finetuning runs that fail with chance-level accuracy. Mixout significantly reduced the number of such bad runs, and this mainly improved the average dev score on each task. In my opinion, your experiment ("cased" vs. "uncased" in terms of mean dev scores) would show that pre-training with the cased input sentences can make the model stable. If this hypothesis is right, then the effectiveness of mixout is less significant for the average score of the "cased" model than for that of the "uncased" one. (I'm not sure, we have not applied mixout to the "cased" model.)
>
> References
> [1] Jacob Devlin, Ming-Wei Chang, Kenton Lee, and Kristina Toutanova. Bert: Pre-training of deep bidirectional transformers for language understanding. arXiv preprint arXiv:1810.04805, 2018.
>
> With regards,
> Cheolhyoung Lee

---

### Decision · Program_Chairs · 2019-12-19

**Decision:**

Accept (Poster)

**Comment:**

This paper presents mixout, a regularization method that stochastically mixes parameters of a pretrained language model and a target language model. Experiments on GLUE show that the proposed technique improves the stability and accuracy of finetuning a pretrained BERT on several downstream tasks.

The paper is well written and the proposed idea is applicable in many settings. The authors have addressed reviewers concerns' during the rebuttal period and all reviewers are now in agreement that this paper should be accepted.

I think this paper would be a good addition to ICLR and recommend to accept it.